# Impact of Genetic Polymorphism on Response to Therapy in Non-Alcoholic Fatty Liver Disease

**DOI:** 10.3390/nu13114077

**Published:** 2021-11-15

**Authors:** José Ignacio Martínez-Montoro, Isabel Cornejo-Pareja, Ana María Gómez-Pérez, Francisco J. Tinahones

**Affiliations:** 1Department of Endocrinology and Nutrition, Virgen de la Victoria University Hospital, 29010 Málaga, Spain; joseimartinezmontoro@gmail.com (J.I.M.-M.); fjtinahones@hotmail.com (F.J.T.); 2Faculty of Medicine, University of Málaga, 29071 Málaga, Spain; 3Instituto de Investigación Biomédica de Málaga (IBIMA), Virgen de la Victoria University Hospital, 29010 Málaga, Spain; 4Spanish Biomedical Research Center in Physiopathology of Obesity and Nutrition (CIBERObn), Instituto de Salud Carlos III, 28029 Madrid, Spain

**Keywords:** non-alcoholic fatty liver disease, gene polymorphism, dietary intervention, gene-nutrient interactions, bariatric surgery, pharmacotherapy

## Abstract

In the last decades, the global prevalence of non-alcoholic fatty liver disease (NAFLD) has reached pandemic proportions with derived major health and socioeconomic consequences; this tendency is expected to be further aggravated in the coming years. Obesity, insulin resistance/type 2 diabetes mellitus, sedentary lifestyle, increased caloric intake and genetic predisposition constitute the main risk factors associated with the development and progression of the disease. Importantly, the interaction between the inherited genetic background and some unhealthy dietary patterns has been postulated to have an essential role in the pathogenesis of NAFLD. Weight loss through lifestyle modifications is considered the cornerstone of the treatment for NAFLD and the inter-individual variability in the response to some dietary approaches may be conditioned by the presence of different single nucleotide polymorphisms. In this review, we summarize the current evidence on the influence of the association between genetic susceptibility and dietary habits in NAFLD pathophysiology, as well as the role of gene polymorphism in the response to lifestyle interventions and the potential interaction between nutritional genomics and other emerging therapies for NAFLD, such as bariatric surgery and several pharmacologic agents.

## 1. Introduction

Non-alcoholic fatty liver disease (NAFLD) has become the leading cause of chronic liver disease worldwide, with an estimated global prevalence of 25% among the adult population [1]. NAFLD comprises the full spectrum of the disease, from simple macrovesicular steatosis to non-alcoholic steatohepatitis (NASH), which is defined by the coexistence of steatosis, hepatocyte ballooning and inflammation with or without fibrosis, presenting an increased risk of progression to cirrhosis, hepatocellular carcinoma and end-stage liver disease [2]. In addition to the important clinical consequences derived from NAFLD, socioeconomic costs of this pathology have been reported to be enormous and the disease burden is expected to continue to increase in the coming years [3].

NAFLD is directly associated with the different components of the metabolic syndrome, including obesity, type 2 diabetes mellitus (T2DM), dyslipidemia and hypertension; in fact, obesity and insulin resistance/T2DM constitute the most common risk factors for NAFLD [4]. Actually, this disease is considered as the hepatic manifestation of the metabolic syndrome and bidirectional relationships between NAFLD and the rest of metabolic complications have been established [5]. Accordingly, environmental factors such as sedentary lifestyle and high-caloric intake play a major part in NAFLD development and progression [6]. However, it is important to note that NAFLD pathogenesis is complex and several factors are involved in the natural history of this pathology. In line, novel environmental modifiers, such as gut microbiota, have been proved to directly influence the course of the disease [7]. Importantly, genome-wide association studies (GWAS) and candidate gene studies have revealed, in the last few years, that NAFLD development, severity and risk of progression are strongly influenced by a number of single-nucleotide polymorphisms (SNPs), including patatin-like phospholipase domain-containing protein 3 (PNPLA3), transmembrane 6 superfamily member 2 (TM6SF2), membrane bound O-acyltransferase domain containing 7 (MBOAT7), glucokinase regulatory protein (GKRP) and hydroxysteroid 17-βdehydrogenase 13 (HSD17B13) as the main genetic determinants of NAFLD [8]. Moreover, the intricate interaction of genetic predisposition and environmental factors, such as nutrition, is considered to play a key role in the pathophysiology of NAFLD [9].

To date, the mainstay of treatment for NAFLD is weight loss [5]. Lifestyle intervention through dietary habits modifications and structured physical activity enables sustained weight loss and the subsequent hepatic fat content reduction and NASH improvement [10]. Importantly, the effect of specific training modalities, such as endurance training, may contribute to NASH improvement [11]. Furthermore, bariatric surgery (BS) has emerged as an effective therapeutic approach to NASH and fibrosis resolution [12]. However, a significant number of patients do not achieve the expected results even after adequate adherence to therapy. Thereby, among other factors, nutrient-gene interaction could explain this inter-individual variability in response to treatment and gene-based personalized therapies may constitute a useful tool in NAFLD treatment. In this article, we review the role of gene polymorphism in the variability in response to therapy in NAFLD, including the interaction between SNPs and dietary interventions, as well as the potential relationships among nutritional genomics, BS and other therapies.

## 2. Nutrigenetics and NAFLD Pathogenesis

In addition to the classical metabolic risk factors for NASH and fibrosis progression, several studies have identified genetic associations with NAFLD susceptibility and severity [13]. The I148M (rs738409 C > G) variant of PNPLA3 (isoleucine to methionine exchange at the amino acid position 148 due to cytosine to guanine transversion in rs738409) is the most important risk mutation related to NAFLD, and it is strongly associated with the development and progression of the disease and also with the response to treatment [14] (Figure 1). PNPLA3 exhibits triacylglycerol lipase and acylglycerol transacylase activity in the hepatocytes and the I148M variant causes loss of function, promoting triglyceride accumulation in the hepatocytes [15]. The frequency of the I148M allele is particularly high in Hispanics (0.49), with lower frequencies in European Americans and African Americans; therefore, this fact may partially explain the differences in NAFLD prevalence among different ethnic groups [16]. On the other hand, TM6SF2 regulates hepatic lipid metabolism and the E165K missense variant impairs very low-density lipoprotein (VLDL) secretion and triggers hepatic lipid accumulation [17], whereas MBOAT7 rs641738 C > T SNP increases risk of NAFLD through the imbalance of phosphatidylinositol species [18]. GKRP rs780094 C > T variant presents a reduced capacity of glucokinase inhibition and consequently enhances glycolysis and glucose uptake by the liver [19]. Conversely, HSD17B13 rs6834314 A > G variant, involved in retinol metabolism, protects against NAFLD progression [20]. Finally, other reported genetic determinants associated with NAFLD include SH2B Adaptor Protein 1 (SH2B1), superoxide dismutase 2 (SOD2), signal transducer and activator of transcription 3 (STAT3), phosphatidylethanolamine-N–methyltransferase (PEMT), apolipoprotein B (APOB) or uncoupling protein 2 (UCP2) [21]. Of note, there are some mitochondria-related SNPs among the NAFLD-associated genetic determinants, since mitochondria dysfunction increases oxidative stress which is closely related to NAFLD pathogenesis [22]. Thus, C47T variant in the mitochondrial enzyme SOD2 is linked to advanced fibrosis in NASH [23], whereas mitochondrial UCP2-866 G > A polymorphism reduces risk of NASH progression [24]. Furthermore, mitochondrial deoxyribonucleic acid (DNA) polymorphism 12361 A > G was associated with increased risk of moderate and severe NAFLD in a Chinese population [25].

Notably, SNP-mediated liver damage only explains a small proportion of NAFLD pathophysiology, and synergistic interaction between these risks variants and the environment are needed to trigger significant alterations [26]. As an example, Smagris et al., showed that PNPLA3 I148M knock in mice developed sucrose diet-dependent hepatic steatosis, but no hepatic alterations were found in chow-fed animals with the mutation [27]. Moreover, a preclinical study revealed that several mitochondrial gene polymorphisms only predisposed to NASH when either a methionine and choline deficient diet or Western-style diet was administrated [28]. Thus, the interaction between nutrients and genetic factors could modulate NAFLD presence and evolution. Additionally, it is also important to bear in mind that nutrition can also give rise to modifications in gene expression through several epigenetic mechanisms, including histone modification, DNA methylation and the regulation of transcription by micro-ribonucleic acids (miRNAs) [29]. These complex pathways are encompassed within the field of nutrigenomics, which constitutes a key element in NAFLD pathogenesis [29]. However, this topic is beyond the scope of this review and we will focus on the influential effect of genetics on response to different nutrients in NAFLD.

### 2.1. Carbohydrates

Dietary carbohydrates, including free sugars, can promote the accumulation of liver fat by increasing intrahepatic triglyceride content [30] and the presence of some SNPs may involve an additive harmful effect. Among them, the most studied nutrient/diet-gene interactions in clinical studies include the I148M variant of PNPLA3. Thus, in a cross-sectional study, Davis et al., reported a positive association between high dietary carbohydrate/sugar consumption and hepatic fat accumulation in Hispanic children with overweight and PNPLA3 GG genotype [31]. Similarly, Nobili et al., found that carriers of this genotype with a high consumption of sweetened beverages presented higher degrees of hepatic steatosis [32]. Furthermore, in a small clinical trial including 14 adolescents, GKRP rs1260326 TT variant increased de novo lipogenesis after glucose overload [33].

Particularly, the consumption of the monosaccharide fructose has been implicated in the development and progression of NAFLD [34]. Fructose triggers hepatic de novo lipogenesis via increasing the levels of lipogenic enzymes and stimulating sterol regulatory element-binding protein (SREBP)-1, and it also inhibits fatty acid oxidation, leading to an increase of reactive oxygen species (ROS) [35]. Thus, an ongoing clinical trial aims to evaluate the impact of fructose intake on liver lipogenesis in subjects with different genetic risk categories for NAFLD [36]. In a case control study, the combination of distinct gene variants related to oxidative stress mechanisms (glutathione S-transferase theta 1-GSTT1, glutathione S-transferase mu 1-GSTM1, sulfotransferase family 1A member 1-SULT1A1, cytochrome P450 2E1-CYP2E1 and cytochrome P450 1A1-CYP1A1) with high fruit/grilled food consumption increased the risk for NAFLD development [37]. In line with these results, previous studies have shown that high fructose diet promotes hepatic steatosis, oxidative stress and inflammation, leading to hepatocyte apoptosis [38]. The pathophysiology of fructose induced-NAFLD via oxidative stress encompasses several mechanisms, such as nonenzymatic reactions of fructose and ROS generation, hepatic phosphate deficiency and the production of harmful metabolites (e.g., methylglyoxal) [39]. In addition, the severity of liver injury by fructose may be mediated by the induced degree of mitochondrial dysfunction and oxidative damage [40]. On the other hand, the hepatic deleterious effects of fructose may be counteracted by some nutrients that prevent oxidative stress and increase the expression of antioxidant defense enzymes [41,42,43,44,45]. Dietary advanced glycation end products compounds found in grilled food have also been postulated to aggravate NAFLD via liver injury induced by chronic oxidative stress, and pharmacological and dietary strategies targeting the implied pathways could help to ameliorate NAFLD [46]. Therefore, the interaction between fructose/grilled food consumption and SNPs involved in oxidative stress may be crucial in NAFLD pathogenesis and resolution.

### 2.2. Lipids

In addition to carbohydrate overfeeding, dietary fat pattern may interrelate with some genotypes. In this sense, Santoro et al., showed that the interaction between PNPLA3 I148M and a high ratio of omega-6/omega-3 polyunsaturated fatty acid (PUFA) intake was associated with higher serum levels of alanine transaminase (ALT) and hepatic fat accumulation [47]. Jones et al., reported that the intake of several dietary types of unsaturated fat, including omega-6, was associated with liver fibrosis by PNPLA3 rs738409 variants [48]. Furthermore, the interaction between SH2B1 rs7359397 T allele and high protein/low fiber and monounsaturated fatty acid (MUFA) consumption may be associated with NAFLD severity [49].

Growing body of evidence supports that disturbances in cholesterol homeostasis contribute to the pathophysiology of NAFLD/NASH [50]. Beyond hepatic accumulation of fatty acids and triglycerides, an increase in free cholesterol deposition in the liver leads to hepatocyte injury [51]. Atherogenic dyslipidemia, a common feature of the metabolic syndrome, may facilitate this fact [52]. Remarkably, high cholesterol atherogenic diets may interact with SNPs involved in cholesterol metabolism. In a study including women that received a high cholesterol Western-type diet, the microsomal triglyceride transfer protein (MTTP)-493 T/T variant was associated with higher fasting levels of plasma cholesterol and higher cholesterol absorption status, whereas these levels decreased to values comparable to G carriers after 3 months of low-fat diet [53]; this variant was related to an increased risk of NAFLD compared with G/G carriers [54]. TM6SF2 C > T polymorphism implies a less atherogenic lipoprotein profile and postprandial cholesterol redistribution from smaller atherogenic lipoprotein subfractions to larger VLDL subfractions in subjects with NAFLD [55]; however, specific interactions with high cholesterol diets remain unexplored. Similarly, SREBP-1c polymorphism is also closely implicated in cholesterol metabolism in NAFLD [56], yet dietary interactions have not been investigated.

### 2.3. Choline Deficiency in NAFLD

Choline is a key nutrient in NAFLD pathogenesis, as its deficiency is closely related to the onset and progression of this disease [57,58]. Susceptibility to choline deficiency and the subsequent increased risk of developing NAFLD may be influenced by specific polymorphisms in genes that regulate choline metabolism, such as PEMT [59,60]. In addition, a study showed that carriers of the 5,10-methylenetetrahydrofolate dehydrogenase (MTHFD)-1958A gene allele were more likely to develop NAFLD on a low-choline diet than non-carriers [61].

In light of the above, nutrient-gene interaction may play a crucial role in NAFLD pathogenesis, although large-scale, long-term prospective clinical studies are needed to corroborate these associations.

## 3. Gene Polymorphism and Response to Lifestyle Interventions in NAFLD

### 3.1. Dietary Changes

Currently, the primary treatment for NAFLD is based on lifestyle modifications, including diet and physical activity to achieve weight loss [62]. The role of nutritional intervention has been demonstrated to be essential for the prevention and management of NAFLD in a number of randomized controlled clinical trials [63,64,65,66,67]. Recent evidence also suggests that the presence of different SNPs combined with some dietary patterns may increase the effect of this approach. In a clinical trial performed within the Fatty Liver in Obesity (FLiO) Study, carriers of T allele of the SH2B1 rs7359397 genetic variant exhibited greater benefits in terms of hepatic health and liver status after two energy-restricted dietary patterns [68]. Interestingly, in a study performed in 140 Japanese patients with biopsy-proven NAFLD, the reduction in liver stiffness measurement after diet therapy for one year was greater among subjects with HSD17B13 rs6834314 GG variant [69]. Previously, in a pilot study conducted by Sevastianova et al., the homozygous subjects for the PNPLA3 rs738409 G allele experienced a more significant decrease in liver fat content in response to a 6-day hypocaloric low carbohydrate diet [70], and a post-hoc analysis of a randomized controlled trial including 154 patients revealed that this genotype was associated with a greater reduction in intrahepatic triglyceride content, body weight and waist-to-hip ratio after a dietitian-led lifestyle program based on a reduced caloric intake for 12 months [71]. Conversely, in a cohort study of 51 children, Koot et al., did not find any relationship between PNPLA3 rs738409 SNP and liver steatosis improvement in a 6-month intensive lifestyle treatment [72], and neither PNPLA3 nor TM6F2 variants were related to NAFLD improvement after a 4-month reduction of caloric intake, although these risk genotypes did not impair the response of dietetic intervention [73]. In addition to the aforementioned SNPs, the Gly385Arg polymorphism in fibroblast growth factor receptor 4 (FGFR4) was not linked to liver fat content or insulin sensitivity in 170 subjects with overweight/obesity at baseline, but it was associated with less decrease in liver fat accumulation and insulin sensitivity under healthy dietary conditions [74]. On the other hand, the presence of the STAT3 rs2293152 G genotype was associated with more beneficial changes after 24-week Mediterranean diet in an open-label study including 44 patients with NAFLD [75].

Thus, although further research is needed, some genetic variants associated with NAFLD development, severity and risk of progression may also confer an enhanced response to dietary intervention, and personalized dietary treatment depending on the presence of specific genetic polymorphism may constitute an attractive approach for NAFLD management. Furthermore, nutritional strategies based on the nutrient-induced insulin output ratio (NIOR) could help to select sensitive SNPs associated with fat and carbohydrate metabolism and design individualized nutrition plans for patients with NAFLD [76].

### 3.2. The Role of Omega-3 PUFA

Omega-3 PUFA supplementation might reduce liver fat, although well designed randomized controlled trials are required to assess their potential role in NAFLD [77,78]. In the last few years, dietary omega-3 PUFA and/or PUFA supplementation has also been related to NAFLD outcomes in the presence of some genetic determinants with mixed results (Table 1). On the one hand, Nobili et al., reported that I148M variant of PNPLA3 led to a decreased response to docosahexaenoic acid (DHA) supplementation in 60 children with NAFLD [79]. Moreover, in the WELCOME trial, the PNPLA3 148M/M genotype was associated with higher liver fat percentage and lower DHA tissue enrichment after 4 g DHA + eicosapentaenoic acid (EPA) supplementation for 15–18 months, although the TM6F2 E167K variant did not show significant associations [80]. Recently, an open-label study showed that short-term omega 3 PUFA intervention (DHA + EPA) did not change liver fat content regardless of the PNPLA3 148M variant [81]. In the EFFECT-I trial, PNPLA3 I148M did not influence the effects of omega-3 PUFA or fenofibrate on liver proton density fat fraction [82]. By contrast, a low omega-6 to omega-3 PUFA ratio diet reduced hepatic fat fraction in a significant higher percentage in the carriers of PNPLA3 148M/M genotype [83], and these results were concordant with those previously reported by Santoro et al. [33]. These findings may be explained by PNPLA3 rs738409 I148M-derived protein decreased ability in hydrolyzing omega-9 PUFA from glycerolipids; being omega-9 PUFA synthetized from omega-6 PUFA, omega-6 overload would increase intrahepatic triglyceride content [84]. Hence, further investigation is needed to elucidate the role of the interaction between omega-3 PUFA and PNPLA3 rs738409 in NAFLD and the study of alternative SNPs may help to find new relationships.

### 3.3. Specific Nutrients

There is a growing interest in the potential benefits of natural supplements in the therapeutic landscape for NAFLD [86] and nutrigenetic approaches in this field may constitute an attractive option. In this sense, Mastiha, a natural product of the Mediterranean basin extracted from the *Pistacia lentiscus* tree, may reduce NASH and fibrosis via its anti-inflammatory, antioxidant and lipid-lowering properties, as well as the restoration of gut microbiota diversity [87,88]. The recent randomized trial MAST4HEALTH assessed the role of nutrigenetic interactions in the modulation of the anti-inflammatory and antioxidant effects of 6-months Mastiha supplementation on NAFLD [89]. In this study, several gene-by-Mastiha interactions were identified, and these associations were linked to levels of cytokines and antioxidant biomarkers after Mastiha treatment, some of them closely related to NAFLD pathogenesis [89]. Silymarin could also be effective in reducing transaminase levels in patients with NAFLD [90], however this effect may be attenuated in PNPLA3 G-allele carriers [91]. On the contrary, although Chia (*Salvia hispanica*), a source of omega-3 PUFA, antioxidants and fiber, may ameliorate NAFLD, no differences in response to this treatment have been found among PNPLA3 different SNPs [92]. In addition, in a pilot trial in subjects with obesity, supplementation with licorice (*Glycyrrhiza glabra*) resulted in significant changes in anthropometric parameters and insulin sensitivity only in those patients with the Pro/Pro SNP of the peroxisome proliferator-activated receptor gamma-2 (PPARγ2) [93]. Thus, given the potential benefits of licorice in NAFLD [94], genetic determinants may explain the variability in response to this nutrient. Folate serum levels may correlate with NASH severity [95], and folic acid supplementation has demonstrated to attenuate hepatic lipid accumulation and inflammation through the restoration of peroxisome proliferator-activated receptor alpha (PPARα), among other mechanisms [96]. Furthermore, the supplementation with folic acid in individuals with the high-risk variant MTHFD 1958A could attenuate signs of choline deficiency [61]. On the other hand, there are a number of nutraceuticals that could exert positive effects on NAFLD; however, their interaction with NAFLD-related SNPs is yet to be studied. In this regard, coenzyme Q10, as an activator of adenosin 5′ monophosphate activated protein kinase (AMPK), has been shown to alleviate NAFLD through the inhibition of lipogenesis and activation of fatty acid oxidation [97]. Paeoniflorin, a peony root component, improved biochemical and histological changes in NAFLD in animal models via insulin-sensitizing and antioxidant effects [98]. Resveratrol, a non-flavonoid phenol derived from grape skins, can attenuate insulin resistance and hepatic oxidative stress in NAFLD [42] and these effects may be mediated by changes in the gut microbiota, an essential component in NAFLD pathophysiology [99]. Supplementation with curcumin, extracted from *Curcuma longa* root, was associated with benefits on NAFLD through the amelioration of insulin resistance and lipid metabolism in both preclinical and clinical studies [100,101]. Berberine, an extract from the genus *Berberis* species, has a role on hepatic lipid metabolism and has been reported to be effective in NAFLD and related metabolic disorders [102]. In view of the foregoing, additional studies including gene-natural antioxidants/food supplements interaction might shed light on NAFLD personalized therapy.

### 3.4. Physical Activity

Physical exercise is one of the cornerstones of NAFLD therapy [103], however available data regarding potential interactions with gene polymorphisms remain scarce. In a case-control study conducted in 1027 Chinese children, physical activity was demonstrated to modulate the effect of PNPLA3 rs738409 variant: proportions of NAFLD increased with the presence of the G-allele only in participants with insufficient physical activity/sedentary behavior [104], and Muto et al., found similar results in a retrospective longitudinal study [105]. With regard to patients with NAFLD, some studies evaluated the impact of lifestyle intervention, including dietary modifications along with physical exercise recommendations [71,72,74] with different results, but the specific physical activity-gene interactions have not been evaluated to date.

## 4. Future Perspectives in NAFLD Treatment: Toward Personalized Therapies?

### 4.1. Bariatric Surgery and NAFLD

BS is considered the most effective treatment to achieve substantial weight loss, thus it constitutes an important therapeutic option for obesity and related comorbidities, including NAFLD [106]. In fact, BS is associated with NASH and fibrosis resolution in a significant number of patients, however a percentage of individuals do not experience enough histopathological improvement after this procedure [12]. Considerably, nutritional genomics play an essential part in personalized bariatric approaches, and the complex crosstalk between these two matters can generate reciprocal influences [107]. Different SNPs involved in the metabolic homeostasis are closely related to BS outcomes and, at the same time, BS induces both genetic and epigenetic modifications that have a major influence on metabolic pathways [108,109].

In this context, there is limited evidence with regard to the impact of gene polymorphism on BS outcomes in patients with NAFLD. In a prospective study including 84 individuals with obesity that underwent BS, PNPLA3 148M variant was associated with increased intrahepatic lipid accumulation before BS, but also with higher reduction of hepatic fat content and weight loss 12 months after the intervention [110]. Conversely, neither TM6F2 nor MBOAT7 showed significant associations [110]. Interestingly, several SNPs have been associated with lower hunger feelings and increased weight loss after BS, while other genetic determinants such as mitochondrial UCP2 have been proved to induce greater energy and carbohydrate intake after Roux-En-Y gastric bypass [111,112]. Hence, genetic determinants for predicting weight loss/regain after BS could be a useful tool to determine the success of this procedure, and NAFLD-related outcomes may be also affected by these SNPs.

### 4.2. Other Therapies

Glucose-lowering agents may be an effective treatment for NAFLD in patients with and without T2DM [62]. Among them, thiazolidinediones have shown several benefits, even in patients with advanced stages of NAFLD [113]. Remarkably, a substudy of 55 participants from a clinical trial to assess long-term efficacy of pioglitazone in NASH, identified SNPs associations with pioglitazone histologic response, including adenosine A1 receptor (ADORA1) rs903361, ATP binding cassette subfamily A member 1 (ABCA1) rs2230806, potassium voltage-gated channel subfamily Q member 1 (KCNQ1) rs2237895, PPAR*γ* rs4135275 and PPAR*γ* rs17817276, among others, and a genetic response score was designed based on the sum of response-associated alleles [114]. In the EFFECT-II study, 84 patients with T2DM and NAFLD were randomly assigned to 10 mg dapagliflozin/4 g omega-3 PUFA/a combination of both/placebo, and an interaction between PNPLA3 I184M (C/C vs. C/G + G/G) and reduction in liver fat content assessed by MRI was found across the active treatment groups [85] (Table 1). Moreover, the G allele carriers had an enhanced response to treatment only in the combined arm, what suggests synergistic effects between therapies in this genotype [85]. Additionally, in a retrospective study with 41 patients with NAFLD and T2DM the response to the dipeptidyl peptidase-4 inhibitor alogliptin was greater in PNPLA3 G-allele carriers [115], albeit in small study conducted in patients with T2DM, PNPLA3 GG genotype was linked to a diminished response to the glucagon-like peptide 1(GLP-1) receptor agonist exenatide in terms of reducing liver fat content [116].

SNPs may also regulate response to Vitamin E treatment in NAFLD. Gene polymorphism of cytochrome P450 4F2 might affect Vitamin E pharmacokinetics and could determine variability in its efficacy, as demonstrated a study with data from the PIVENS and TONIC clinical trials [117]. However, a retrospective study showed that liver stiffness reduction in patients with NAFLD taking Vitamin E was not influenced by PNPLA3 genotypes [118]. On the other hand, several genetic predictors of response to obeticholic acid in patients with NASH were identified in a pilot GWAS study, with the CELA3B rs75508464 variant with the most significant effect on NASH resolution [119].

Finally, the restoration of gut microbiota through the use of probiotics/symbiotics may constitute an interesting therapeutic approach in NAFLD [120]. Gut microbiota dysbiosis has a central role in NAFLD pathogenesis [121] and microbiota-derived metabolites (bile acids, short-chain fatty acids, branched-chain amino acids, etc.) are also important modulators of the disease [122]. Gut microbiome based metagenomic signature could be useful for the diagnosis of advanced stages of NAFLD [123], and gut microbiota-miRNA interactions have been reported to impact on NAFLD pathophysiology [124]. In animal models, the combination of blueberry juice and probiotics has been proved to improve NASH via increasing PPAR*α* and reducing the levels of *SREBP-1* and *PNPLA3* [125]. Nevertheless, the potential interactions between probiotics/symbiotics and specific SNPs remain unknown.

## 5. Concluding Remarks

NAFLD is the most common cause of chronic liver disease globally and involves important clinical and socioeconomic implications. Gene polymorphism-nutrient interaction plays a central role in NAFLD pathogenesis and the effectiveness of lifestyle interventions, including dietary modifications, seems to be also modulated by different genetic determinants. In this review, a number of SNPs closely related to pathways involved in NAFLD (e.g., mitochondrial dysfunction, oxidative stress, lipid metabolism) and their interaction with both proven effective dietary patterns/food components and promising novel nutraceuticals for the treatment of NAFLD have been described. Since the variability in response to therapy in NAFLD may be explained by this fact, the assessment of key NAFLD-related SNPs in interventional studies should be considered. Moreover, gene- based personalized diet therapy may constitute a helpful option for the management of NAFLD, although more well-conducted large-scale, long-term trials assessing the influence of SNPs on the response to specific dietary approaches (e.g., Mediterranean diet, low-carbohydrate diet, intermittent fasting) and single nutrients are needed. Furthermore, these effects should be also evaluated in advanced stages of NAFLD. Finally, this review includes an integrative view of the emerging therapies and targets for NAFLD, pointing out the potential interplay between nutritional genomics, physical exercise, BS, pharmacotherapy and the gut microbiota in this pathology. Although recent studies have shown promising results in this regard, further investigation is warranted to determine its impact.

## Figures and Tables

**Figure 1 nutrients-13-04077-f001:**
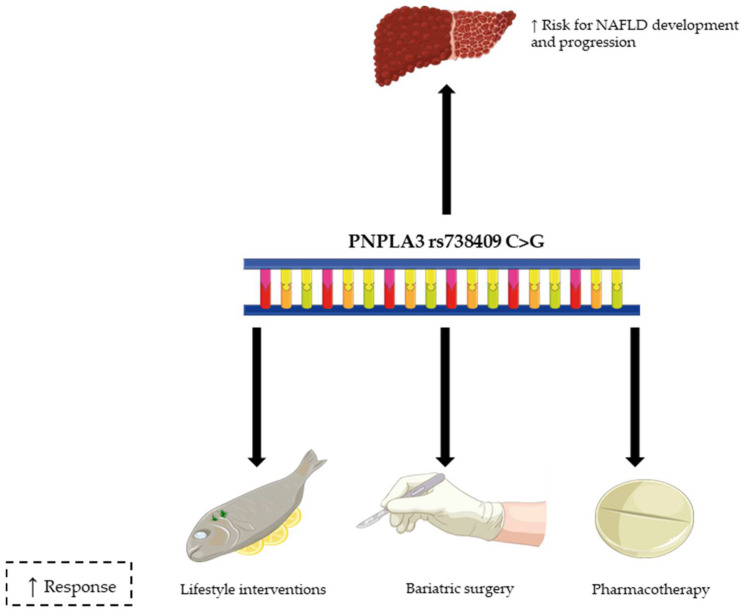
The role of PNPLA3 rs738409 C > G variant in NAFLD. PNPLA3 I148M is associated with NAFLD development and progression, and the interplay between this variant and environmental factors, including dietary habits, seems to be crucial in the pathophysiology of the disease. PNPLA3 I148M presence may also be related to an increased response to lifestyle interventions, bariatric surgery and certain types of therapeutic agents, such as the combination of sodium-glucose cotransporter 2 inhibitors and polyunsaturated fatty acid (PUFA). PNPL3: patatin-like phospholipase domain-containing protein 3; I148M (rs738409 C > G): isoleucine to methionine exchange at the amino acid position 148 due to cytosine to guanine transversion in rs738409); NAFLD: non-alcoholic fatty liver disease.

**Table 1 nutrients-13-04077-t001:** Clinical studies assessing the role of the interaction between omega-3 PUFA and PNPLA3 rs738409 in NAFLD.

Study	Design (Sample Size)	Intervention (Time)	Result
Santoro et al., 2012 [47]	Cross-sectional study (127)	-	Higher HFF% and ALT levels in 148M/M variant presenting high dietary n-6/n-3 PUFA consumption
Nobili et al., 2013 [79]	RCT (60)	DHA 250–500 mg/day (24 months)	Lower response (steatosis) in I148M variant
Scorletti et al., 2015 [80]	RCT (85)	DHA + EPA 4 g/day (15–18 months)	Increased end of study liver fat % in 148M/M variant
Eriksson et al., 2018 [85]	RCT (84)	10 mg dapagliflozin/4 g n-3 PUFA/both (12 weeks)	Combined treatment induced greater response (PDFF) in I148M variant; n-3 PUFA treatment induced decreased response (PDFF) in I148M variant
Oscarsson et al., 2018 [82]	RCT (78)	200 mg fenofibrate/4 g n-3 PUFA (12 weeks)	No influence of I148M on the effects of n-3 PUFA supplementation (PDFF)
Kuttner et al., 2019 [81]	Open-label trial (20)	4 g n-3 PUFA (4 weeks)	No changes in transient elastography (CAP used to quantify liver fat) neither in the control group nor I148M
Van Name et al., 2020 [83]	Single-arm unblinded trial (20)	Low n-6/n-3 PUFA ratio (4:1) normocaloric diet (12 weeks)	Significant HFF% reduction in the 148M/M group

HFF%: hepatic fat fraction (%); ALT: alanine aminotransferase; n-6/n-3: omega-6/omega-3 ratio; PUFA: polyunsaturated fatty acids; RCT: randomized clinical trial; DHA: docosahexaenoic acid; EPA: eicosapentaenoic acid; PDFF: proton density fat fraction; CAP: controlled attenuation parameter.

## Data Availability

Not applicable.

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
