# Peer review of "Impact of Genetic Polymorphism on Response to Therapy in Non-Alcoholic Fatty Liver Disease"

_nutrients, 2021, doi:10.3390/nu13114077_

Round 1
Reviewer 1 Report
The author of this manuscript reviews the association of SNPs in NAFLD. But, The scope and approach of this paper is less novel. Authors should emphasize the original and creative perspective found by the authors in "5. concluding remarks." Also, the lack of mention of cholesterol in discussing NAFLD, SNPs and nutrition feels unsatisfactory as a review, especially as nutrients.
minor point
・L98 Delete unnecessary space.
"NAFLD development and progression"
・L455-457 Correction of reference.
"vascular Risk. :1920–9."
→J. Nutr 2019; 149(11):1920-1929. doi:10.1093/jn/nxz147.
Author Response
Dear Editor and Reviewers,
We would like to thank you very much for your constructive comments and suggestions which have undoubtedly helped us to improve our manuscript.
We have taken these comments and suggestions into consideration and have revised the paper accordingly. We have made all possible efforts to respond to each of the reviewers’ comments and have edited the manuscript where we were able to address the reviewers’ suggestions fully.
We have provided the replies to the comments in the following section and have highlighted changes in the manuscript in a red font.
We hope that our revised manuscript may now be found acceptable for publication in the journal. Nevertheless, we are of course willing to revise it further according to any other suggestions or concerns raised by the Editor or the Reviewers.
Yours faithfully,
The authors
Reviewer Comments to Authors
Reviewer 1:
Comment 1: The author of this manuscript reviews the association of SNPs in NAFLD. But, the scope and approach of this paper is less novel. Authors should emphasize the original and creative perspective found by the authors in "5. concluding remarks."
Response 1: We would like to thank the reviewer for his/her commentary.
Following the reviewer´s advice, we have extensively modified the section “5. Concluding Remarks” in order to highlight the original perspective of this review (lines 403-410; 414-418):
“In this review, a number of SNPs closely related to pathways involved in NAFLD (e.g. mitochondrial dysfunction, oxidative stress, lipid metabolism) and their interaction with both proven effective dietary patterns/food components and promising novel nutraceuticals for the treatment of NAFLD have been described. Because the variability in response to therapy in NAFLD may be explained by this fact, the assessment of key NAFLD-related SNPs in interventional studies should be considered. Moreover, gene- based personalized diet therapy may constitute a helpful option for the management of NAFLD”.
“Finally, this review includes an integrative view of the emerging therapies and targets for NAFLD, pointing out the potential interplay between nutritional genomics, BS, pharmacotherapy and the gut microbiota in this pathology. Although recent studies have shown promising results in this regard, further investigation is warranted to determine its impact”.
Comment 2: Also, the lack of mention of cholesterol in discussing NAFLD, SNPs and nutrition feels unsatisfactory as a review, especially as nutrients.
Response 2: We thank the reviewer for his/her apt observation.
Following the reviewer´s suggestion, we have added a section with regard to cholesterol homeostasis, NAFLD, diet and SNPs (2.2 Lipids, lines 182-198):
“Growing body of evidence supports that disturbances in cholesterol homeostasis contribute to the pathophysiology of NAFLD/NASH (50). Beyond hepatic accumulation of fatty acids and triglycerides, an increase in free cholesterol deposition in the liver leads to hepatocyte injury (51). Atherogenic dyslipidemia, a common feature of the metabolic syndrome, may facilitate this fact (52). Remarkably, high cholesterol atherogenic diets may interact with SNPs involved in cholesterol metabolism. In a study including women that received a high cholesterol Western-type diet, the microsomal triglyceride transfer protein (MTTP)-493 T/T variant was associated with higher fasting levels of plasma cholesterol and higher cholesterol absorption status, whereas these levels decreased to values comparable to G carriers after 3 months of low fat diet (53); this variant was related to an increased risk of NAFLD compared with G/G carriers (54). TM6SF2 C>T polymorphism implies a less atherogenic lipoprotein profile and postprandial cholesterol redistribution from smaller atherogenic lipoprotein subfractions to larger VLDL subfractions in subjects with NAFLD (55); however, specific interactions with high cholesterol diets remain unexplored. Similarly, SREBP 1c polymorphism is also closely implicated in cholesterol metabolism in NAFLD (56), yet dietary interactions have not been investigated”.
Comment 3: minor point・L98 Delete unnecessary space. "NAFLD development and progression" ・L455-457 Correction of reference.
"vascular Risk. :1920–9."→J. Nutr 2019; 149(11):1920-1929. doi:10.1093/jn/nxz147
Response 3: Thank you very much for your appreciations.
Unnecessary space has been deleted (line 112) and reference 63 has been corrected (line 595).
Final comments
Once again, we are really grateful for the opportunity to revise and improve our manuscript. Indeed, we believe that with all suggested changes and modifications the manuscript has significantly improved. Thanks for your feedback.
We hope that our revised manuscript may now be considered acceptable for publication in the journal. Nevertheless, we are of course willing to revise it further according to any other suggestions or concerns raised by the Editor or the Reviewers.
Yours faithfully,
The authors
Reviewer 2 Report
Relevant review.
Please add a section where food/food components known to be beneficial upon NAFLD have not had their interaction with NAFLD-related SNPs studied yet.
Are there any mitochondria-related SNPs relevant in NAFLD? Mitochondria dysfunction increases oxidative stress and oxidative stress contributes to NAFLD.
Please read the following carefully.
- Comments regarding scientific content and references cited.
- ‘To date, the mainstay of treatment for NAFLD is weight loss (5). Lifestyle intervention through dietary habits modifications and structured physical activity enables sustained weight loss and the subsequent hepatic fat content reduction and NASH improvement (10).’
Reviewer. The following reference should be included as the authors show, in an animal model, that physical exercise (and, most importantly, the type of physical exercise) improves NASH.
Gonçalves IO et al. Physical exercise antagonizes clinical and anatomical features characterizing Lieber-DeCarli diet-induced obesity and related metabolic disorders. Clin Nutr. 2015 Apr;34(2):241-7.
- ‘In a case control study, the combination different gene variants related to oxidative stress mechanisms (GSTT1, GSTM1, SULT1A1, CYP2E1 and CYP1A1) with high fruit/grilled food consumption increased NAFLD development risk (31).’
Reviewer. Please include the rationale bridging high fructose consumption, high oxidative stress and NAFLD (a). The same for metabolization of advanced glycation end products compounds found in grilled food.
- The following references should be useful in the requested discussion; so, their inclusion must be considered.
Cardoso RR et al. Kombuchas from green and black teas reduce oxidative stress, liver steatosis and inflammation, and improve glucose metabolism in Wistar rats fed a high-fat high-fructose diet. Food Funct. 2021 Oct 7. doi: 10.1039/d1fo02106k.
García-Berumen CI et al. The severity of rat liver injury by fructose and high fat depends on the degree of respiratory dysfunction and oxidative stress induced in mitochondria. Lipids Health Dis. 2019 Mar 30;18(1):78.
Choi Y, Abdelmegeed MA, Song BJ. Diet high in fructose promotes liver steatosis and hepatocyte apoptosis in C57BL/6J female mice: Role of disturbed lipid homeostasis and increased oxidative stress. Food Chem Toxicol. 2017 May;103:111-121.
Jegatheesan P, De Bandt JP. Fructose and NAFLD: The Multifaceted Aspects of Fructose Metabolism. Nutrients. 2017 Mar 3;9(3):230.
Bagul PK et al. Attenuation of insulin resistance, metabolic syndrome and hepatic oxidative stress by resveratrol in fructose-fed rats. Pharmacological Research. 2012;66(3):260–268.
Suwannaphet W et al. Preventive effect of grape seed extract against high-fructose diet-induced insulin resistance and oxidative stress in rats. Food and Chemical Toxicology. 2010;48(7):1853–1857.
Tsai H-Y, Wu L-Y, Hwang LS. Effect of a proanthocyanidin-rich extract from longan flower on markers of metabolic syndrome in fructose-fed rats. Journal of Agricultural and Food Chemistry. 2008;56(22):11018–11024.
Polizio AH et al. Behaviour of the anti-oxidant defence system and heme oxygenase-1 protein expression in fructose-hypertensive rats. Clinical and Experimental Pharmacology and Physiology. 2006;33(8):734–739.
- ‘Fructose triggers hepatic de novo lipogenesis via increasing the levels of lipogenic enzymes and stimulating sterol regulatory element-binding protein (SREBP)-1, and it also inhibits fatty acid oxidation, leading to an increase of reactive oxygen species (29). Intriguingly, an ongoing clinical trial aims to evaluate the impact of fructose intake on liver lipogenesis ins subjects with different genetic risk categories for NAFLD (30).’
Reviewer. Why was the connector ‘intriguingly’ used? The mentioned ongoing clinical trial is in line with the information given previously.
[A typo can be seen in red.]
- Details and formatting corrections needed.
- Please carefully read the entire manuscript and correct typos; also, reduce and/or introduce spacing between words, and between words and punctuation.
Examples:
. ‘Similarly, Nobili et al …’
. ‘… adolescents, GKRP rs1260326 TT variant increased de novo lipogenesis after glucose overload (27).’
. ‘PNPLA3 rs738409 G allele experienced a more significant dicrease in liver fat content in ...’
. ‘ … the Gly385Arg polymorphism in fibroblast growth factor receptor 4 (FGFR4) was not linked to liver fat content or insulin sensitivity in 170 subjects with overweight/obesity at baseline, but it was associated with less decrease in liver fat accumulation and insulin sensitivity under healthy dietary conditions (52).’
. ‘These findings may be explained by PNPLA3 rs738409 I148M-derived protein decreased ability in hydrolilyzing omega-9 PUFA, synthetized from omega-6 PUFA, therefore omega-6 overload would increase intrahepatic triglyceride content (62).’
. ‘… albeit in small study conducted in patients with T2DM, PNPlA3 GG genotype was linked to a dismished reponse to the the glucagon-like peptide 1 (GLP-1) receptor agonist exenatide in terms of reducing liver fat content (88).’
. ‘Finally, the resotaration of gut microbiota through the use of probiotics/symbiotics …’
- Reviewer. Explain the following abbreviations the first time they are used: GSTT1, GSTM1, SULT1A1, CYP2E1, CYP1A1, SH2B1, SOD2, STAT3, PEMT, APOB and UCP2.
- ‘In this sense, Santoro et al showed that the interaction between PNPLA3 I148M and a high ratio of omega-6/omega-3 polyunsaturated fatty acid (PUFA) intake was associated with higher levels of alanine transaminase (ALT) and hepatic fat accumulation (32).’
Reviewer. ALT levels in liver or in circulation/blood? Please clarify.
Reviewer. EPA (and its explanation) is missing at the end of Table 1.
Reviewer. Eliminate italic formatting in lines 246 to 253 and 326 to 328. Within the former text, two typos can be found: ‘potnetial’ and ‘Furhtermore’.
Author Response
Dear Editor and Reviewers,
We would like to thank you very much for your constructive comments and suggestions which have undoubtedly helped us to improve our manuscript.
We have taken these comments and suggestions into consideration and have revised the paper accordingly. We have made all possible efforts to respond to each of the reviewers’ comments and have edited the manuscript where we were able to address the reviewers’ suggestions fully.
We have provided the replies to the comments in the following section and have highlighted changes in the manuscript in a red font.
We hope that our revised manuscript may now be found acceptable for publication in the journal. Nevertheless, we are of course willing to revise it further according to any other suggestions or concerns raised by the Editor or the Reviewers.
Yours faithfully,
The authors
Reviewer Comments to Authors
Reviewer 2
Relevant review:
Comment 1: Please add a section where food/food components known to be beneficial upon NAFLD have not had their interaction with NAFLD-related SNPs studied yet.
Response 1: Thank you very much for your helpful suggestion.
Following the reviewer´s advice, we have added a section regarding to this topic (3.3 Specific nutrients, lines 303-318):
“On the other hand, there are a number of nutraceuticals that could exert positive effects on NAFLD; however, their interaction with NAFLD-related SNPs is yet to be studied. In this regard, coenzyme Q10, as an activator of adenosin 5’ monophospate activated protein kinase (AMPK), has been shown to alleviate NAFLD through the inhibition of lipogenesis and activation of fatty acid oxidation (97). Paeoniflorin, a peony root component, improved biochemical and histological changes in NAFLD in animal models via insulin-sensitizing and antioxidant effects (98). Resveratrol, a non-flavonoid phenol derived from grape skins, can attenuate insulin resistance and hepatic oxidative stress in NAFLD (42) and these effects may be mediated by changes in the gut microbiota, an essential component in NAFLD pathophysiology (99). Supplementation with curcumin, extracted from Curcuma longa root, was associated with benefits on NAFLD through the amelioration of insulin resistance and lipid metabolism in both preclinical and clinical studies (100,101). Berberine, an extract from the genus Berberis species, has a role on hepatic lipid metabolism and has been reported to be effective in NAFLD and related metabolic disorders (102)”.
Comment 2: Are there any mitochondria-related SNPs relevant in NAFLD? Mitochondria dysfunction increases oxidative stress and oxidative stress contributes to NAFLD.
Response 2: We thank the reviewer for his/her remark.
Following the suggestion of the reviewer, we have added some commentaries about mitochondria-related SNPs relevant in NAFLD (Lines 104-110 and 124-126):
“Of note, there are some mitochondria-related SNPs among these genetic determinants, since mitochondria dysfunction increases oxidative stress, which is closely related to NAFLD pathogenesis (22). Thus, C47T variant in the mitochondrial enzyme SOD2 is linked to advanced fibrosis in NASH (23), whereas mitochondrial UCP2-866 G>A polymorphism reduces risk of NASH progression (24). Besides, mitochondrial deoxyribonucleic acid (DNA) polymorphism 12361 A>G was associated with increased risk of moderate and severe NAFLD in a Chinese population (25)”.
“Moreover, a preclinical study revealed that several mitochondrial gene polymorphisms only predisposed to NASH when either a methionine and choline deficient diet or Western-style diet was administrated (28)”.
Comments regarding scientific content and references cited
Comment 1: ‘To date, the mainstay of treatment for NAFLD is weight loss (5). Lifestyle intervention through dietary habits modifications and structured physical activity enables sustained weight loss and the subsequent hepatic fat content reduction and NASH improvement (10)’. The following reference should be included as the authors show, in an animal model, that physical exercise (and, most importantly, the type of physical exercise) improves NASH. Gonçalves IO et al. Physical exercise antagonizes clinical and anatomical features characterizing Lieber-DeCarli diet-induced obesity and related metabolic disorders. Clin Nutr. 2015 Apr;34(2):241-7.
Response 1: We would like to thank the reviewer for his/her recommendation.
As suggested, we have included this reference in the pertinent section (lines 70-71):
“Importantly, the effect of specific training modalities, such as endurance training, may contribute to NASH improvement (11)”.
Comment 2: ‘In a case control study, the combination different gene variants related to oxidative stress mechanisms (GSTT1, GSTM1, SULT1A1, CYP2E1 and CYP1A1) with high fruit/grilled food consumption increased NAFLD development risk (31).’ Please include the rationale bridging high fructose consumption, high oxidative stress and NAFLD (a). The same for metabolization of advanced glycation end products compounds found in grilled food. The following references should be useful in the requested discussion; so, their inclusion must be considered.
Response 2: We thank the reviewer for his/her practical advice.
As indicated, a rationale bridging high fructose consumption/ metabolization of advanced glycation end products compounds found in grilled food, high oxidative stress and NAFLD has been added, including the recommended references (lines 156-171):
“In line with these results, previous studies have shown that high fructose diet promotes hepatic steatosis, oxidative stress and inflammation, leading to hepatocyte apoptosis (38). The pathophysiology of fructose induced-NAFLD via oxidative stress encompasses several mechanisms, such as nonenzymatic reactions of fructose and ROS generation, hepatic phosphate deficiency and the production of harmful metabolites (e.g. methylglyoxal) (39). Also, the severity of liver injury by fructose may be mediated by the induced degree of mitochondrial dysfunction and oxidative damage (40). On the other hand, the hepatic deleterious effects of fructose may be counteracted by some nutrients that prevent oxidative stress and the expression of anti-oxidant defence enzymes (41–45). Dietary advanced glycation end products compounds found in grilled food have also been postulated to aggravate NAFLD via liver injury induced by chronic oxidative stress, and pharmacological and dietary strategies targeting the implied pathways could help to ameliorate NAFLD (46). Therefore, the interaction between fructose/grilled food consumption and SNPs involved in oxidative stress may be crucial in NAFLD pathogenesis and resolution”.
Comment 3: ‘Fructose triggers hepatic de novo lipogenesis via increasing the levels of lipogenic enzymes and stimulating sterol regulatory element-binding protein (SREBP)-1, and it also inhibits fatty acid oxidation, leading to an increase of reactive oxygen species (29). Intriguingly, an ongoing clinical trial aims to evaluate the impact of fructose intake on liver lipogenesis ins subjects with different genetic risk categories for NAFLD (30).’ Why was the connector ‘intriguingly’ used? The mentioned ongoing clinical trial is in line with the information given previously.
Response 2: Thank you for your appreciation.
The connector “intriguingly” has been replaced (line 150).
Details and formatting corrections needed
Comment 1: Please carefully read the entire manuscript and correct typos; also, reduce and/or introduce spacing between words, and between words and punctuation.
Response 1: Thanks for your commentary
We have extensively revised the manuscript in order to correct typos and spacing errors.
Comment 2: Explain the following abbreviations the first time they are used: GSTT1, GSTM1, SULT1A1, CYP2E1, CYP1A1, SH2B1, SOD2, STAT3, PEMT, APOB and UCP2.
Response 2: We thank the reviewer for his/her observation.
As indicated, the above abbreviations have been explained the first time they are used.
Comment 3: ‘In this sense, Santoro et al showed that the interaction between PNPLA3 I148M and a high ratio of omega-6/omega-3 polyunsaturated fatty acid (PUFA) intake was associated with higher levels of alanine transaminase (ALT) and hepatic fat accumulation (32).’ ALT levels in liver or in circulation/blood? Please clarify.
Response 3: Thank you very much for your appreciation.
We have clarified this point (line 176).
Comment 4: EPA (and its explanation) is missing at the end of Table 1
Response 4: Thanks for your remark.
EPA (and its explanation) has been added at the end of Table 1.
Comment 5: Eliminate italic formatting in lines 246 to 253 and 326 to 328. Within the former text, two typos can be found: ‘potnetial’ and ‘Furhtermore’.
Response 5: Thank you for your commentary.
Formatting has been eliminated in the indicated lines. We have also corrected the mentioned typos.
Final comments
Once again, we are really grateful for the opportunity to revise and improve our manuscript. Indeed, we believe that with all suggested changes and modifications the manuscript has significantly improved. Thanks for your feedback.
We hope that our revised manuscript may now be considered acceptable for publication in the journal. Nevertheless, we are of course willing to revise it further according to any other suggestions or concerns raised by the Editor or the Reviewers.
Yours faithfully,
The authors

Round 2
Reviewer 1 Report
I recommend that it be accepted for publication.
I wish the authors all the best in their future endeavors.
Author Response
Dear Editor and Reviewers,
One more time, we would like to express our sincere gratitude for all your constructive comments and recommendations. We really appreciate your help and advice, which have allowed us to improve our manuscript.
Yours sincerely,
The authors.
Reviewer 2 Report
Please find the review comments below.
Scientific comments.
- The following sentence needs to be corrected in terms of PNPLA3 enzymatic activity: there is no hydrolysis of PUFA but of bonds between PUFA and a ‘skeleton’ (that can be glycerol; and in this case PNPLA3 hydrolyses glycerolipids/triglycerides).
These findings may be explained by PNPLA3 rs738409 I148M-derived protein decreased ability in hydrolyzing omega-9 PUFA from glycerolipids/from triglycerides (???); being omega-9 PUFA synthetized from omega-6 PUFA, omega-6 overload would increase intrahepatic triglyceride content (84).
- Should physical exercise be included in this sentence from the conclusion section of the review?
Finally, this review includes an integrative view of the emerging therapies and targets for NAFLD, pointing out the potential interplay between nutritional genomics, BS, pharmacotherapy and the gut microbiota in this pathology.
Minor English language corrections that need to be done, as well as some formatting adjustments.
- In the last decades, the global prevalence of non-alcoholic fatty liver disease (NAFLD) has reached pandemic proportions with derived major health and socioeconomic consequences; this tendency is expected to be further aggravated in the coming years.
- … with an estimated global prevalence of 25% among the adult population (1).
- In line, novel environmental modifiers, such as gut microbiota, have been proved to directly influence the course of the disease (7).
- Importantly, genome-wide association studies (GWAS) and candidate gene studies have revealed, in the last few years, that NAFLD development, severity and risk of progression are strongly influenced by a number of single-nucleotide polymorphisms (SNPs), ….
- Moreover, the intricate interaction of genetic predisposition and environmental factors, such as nutrition, is considered to play a key role in the pathophysiology of NAFLD (9).
- The frequency of the I148M allele is particularly high in Hispanics (0.49), with lower frequencies in European Americans and African Americans; therefore this fact may partially explain the differences in NAFLD prevalence among different ethnic groups (16).
- Of note, there are some mitochondria-related SNPs among the NAFLD-associated genetic determinants, since mitochondria dysfunction increases oxidative stress which is closely related to NAFLD pathogenesis (22).
- Notably, SNP-mediated liver damage only explains a small proportion of NAFLD pathophysiology, and ….
- ….. with high fruit/grilled food consumption increased the risk for NAFLD development (37).
- On the other hand, the hepatic deleterious effects of fructose may be counteracted by some nutrients that prevent oxidative stress and increase the expression of antioxidant defence enzymes (41–45).
- Similarly, SREBP-1c polymorphism is also closely implicated in cholesterol metabolism in NAFLD ….
- …. of NAFLD in a number of randomized controlled clinical trials (63–67).
- Conversely, in a cohort study of 51 children, …
- …. be attenuated in PNPLA3 G-allele carriers (91).
- … as an activator of adenosine 5’ monophosphate activated protein kinase (AMPK), ….
- Physical exercise is one of the cornerstones of NAFLD therapy (103), however available data regarding potential interactions with gene polymorphims remain scarce.
- …. SNP of the peroxisome proliferator-activated receptor gamma-2 (PPARγ2) (93).
- …. microbiota-miRNA interactions have been reported to impact on NAFLD pathophysiology (124).
- …. via increasing PPARα and reducing the levels of SREBP-1 and PNPLA3 (125).
Author Response
Dear Editor and Reviewers,
One more time, we would like to express our sincere gratitude for all your constructive comments and recommendations. We really appreciate your help and advice, which have allowed us to improve our manuscript.
We have provided the replies to Reviewer 2’s comments in the following section and have highlighted changes in the manuscript in a red font.
We hope that our revised manuscript may now be considered acceptable for publication in the journal. Nevertheless, we are of course willing to revise it further according to any other suggestions or concerns raised by the Editor or the Reviewers.
Yours sincerely,
The authors.
Final comments
Once again, we are really grateful for the opportunity to revise and improve our manuscript.
We hope that our revised manuscript may now be considered acceptable for publication in the journal. Nevertheless, we are of course willing to revise it further according to any other suggestions or concerns raised by the Editor or the Reviewers.
Yours faithfully,
The authors.
